# Characteristics of Individuals Who Chose to Participate in a Preceptor Continuing Professional Development Program

**DOI:** 10.3390/pharmacy8030121

**Published:** 2020-07-20

**Authors:** Shweta Shah, Raymond Chojnacki, Jodi Meyer, Amanda Margolis

**Affiliations:** School of Pharmacy, University of Wisconsin-Madison, Madison, WI 53705, USA; srshah6@wisc.edu (S.S.); rchojnacki@wisc.edu (R.C.); jmeyer33@wisc.edu (J.M.)

**Keywords:** preceptors, continuing professional development, experiential education, pharmacy

## Abstract

An online program for pharmacy preceptors to improve their clinical teaching using continuing professional development (CPD) was launched in 2017. While 491 preceptors participated in the CPD program, only 35% of potential participants completed this voluntary program. A secondary data analysis was undertaken to determine the characteristics of preceptors who completed the program and identify ways to target program advertising for those who did not complete it. Residency-trained preceptors were more likely to complete the CPD program compared with those without residency training (45% and 37%, respectively; *p* = 0.011). This may be due to the inclusion of CPD in residency accreditation standards. To improve completion of the CPD program by preceptors without residencies, a brief introduction to CPD, a statement of benefits, and use of a personalized plan should be included in advertisements. Preceptors teaching more experiential students were more likely to complete the CPD program (*p* < 0.001 for introductory and advanced experiences). To encourage preceptors with less students to participate, the CPD program should be advertised year-round to allow preceptors to complete the training when it is most relevant to their precepting schedule. Future directions include the monitoring of changes in CPD program participation rates following changes in advertisements and exploring other motivations for program completion such as collaborations with employers.

## 1. Introduction

Continuing professional development (CPD) is a formal, individualized process that encourages lifelong learning and consists of the following steps: reflect, plan, learn, apply, and evaluate [1]. Most importantly, the cycle promotes the incorporation of behavior change through the apply step. The CPD cycle also promotes the recording and reviewing of the reflect, plan, learn, and evaluate steps to facilitate the successful completion of self-identified learning goals. Pharmacists utilizing CPD have been shown to develop formal learning plans, set specific learning objectives, and achieve their learning objectives more often than colleagues not trained in the CPD process [2,3]. In fact, pharmacists are specifically encouraged by the American Association of Colleges of Pharmacy to utilize CPD to foster continued clinical and professional growth [4]. Additionally, the Accreditation Council for Pharmacy Education (ACPE) recognizes the value of CPD for pharmacy preceptors by requiring schools of pharmacy to “foster the professional development of [their] preceptors” [5] in Standard 20.3. Delivering valuable and accessible CPD programs to preceptors may be an important way to both develop their skills as preceptors and improve the learning outcomes of pharmacy students, who spend at least 30% of their Doctor of Pharmacy curriculum at practice sites [5]. 

The CPD process has been shown to be an effective and well-received learning tool for pharmacy preceptors to enhance their precepting skills [6,7]. Therefore, Margolis and colleagues offered a free online optional CPD program for preceptors with associated continuing education (CE) credits [7]. The CPD program started with a presentation which introduced the CPD process and then encouraged preceptors to review their feedback from students, reflect on their clinical teaching behaviors, and develop a CPD learning plan with SMART objectives to improve their precepting. Margolis et al. found that 35% of preceptors (491/1400) completed the CPD for a preceptor program and submitted goals to improve their precepting skills. The positive response to this activity suggests that preceptors are receptive to learning about CPD and applying it to their role as preceptors. Additionally, 90.5% of preceptors completing the continuing education CE evaluation from the CPD program indicated they felt they could apply most or all that was learned and over 90% found the materials to be both useful and effective (Margolis, unpublished data, 2018). However, despite a large number of preceptors who completed the program and the perceived value of the program, there was still a large portion of preceptors who did not complete the CPD activity. This low completion rate is also consistent with other pharmacist CPD programs. McConnell and colleagues reported a 30.5% enrollment rate for a live CPD workshop and Tofade and colleagues reported 18% of 236 survey respondents completed at least one online CPD webcast and only 5% of preceptors completed all three CPD webcasts advertised and available for free [6,8]. Understanding the characteristics of this group of preceptors is important for increasing the reach of CPD programs administered by schools and colleges of pharmacy. The objective of this evaluation was to analyze the characteristics of pharmacy preceptors who completed the previously administered online CPD for a preceptor program in order to identify ways to develop targeted messaging and adapt future CPD activities to target characteristics of those who did not complete the CPD program. 

## 2. Materials and Methods 

This secondary data analysis builds off of a previously published cross-sectional evaluation of pharmacy preceptor fidelity to the creation of CPD plans which was conducted at the University of Wisconsin-Madison School of Pharmacy (UW-Madison SOP) [7]. Briefly, this online preceptor CPD program described (1) the CPD process, (2) how CPD differs from CE, and (3) how CPD can be used to improve precepting skills to facilitate student learning. The program was available free of charge to all UW-Madison SOP preceptors. After preceptors completed an informational online PowerPoint presentation, the school made available a variety of preceptor development resources to help facilitate their individual preceptor development plan. Preceptors completed a guided worksheet to assist in developing their CPD plan. Questions on the worksheet focused on recalling what was done well the previous academic year and what areas the preceptor would like to improve on. Then, the preceptors were prompted to write one SMART learning objective and develop a learning plan. The questions, SMART objectives, and plan were submitted via a survey quiz. All content focused on preceptors improving as clinical teachers. 

Participants were able to collect 0.75 CE credits upon completion of the training and answering seven multiple choice and true false questions. The preceptor CPD program was advertised via announcement emails to all school of pharmacy preceptors and a two-month time window was given for completion after the final reminder email. 

### 2.1. Data Collection and Management

For the present analysis to determine characteristics of preceptors who completed the CPD program, there were two independent data sources used, (1) a clerkship administrative database which included preceptor characteristics and (2) the CPD program completion database from the learning management software. The databases were cleaned separately before compiling them for analysis. A total of 1837 cases in the preceptor database represented the total number of pharmacy preceptors at various sites at the time of download. The cases present in either dataset that were unable to merge were excluded from the analysis on account of incomplete information to conduct analysis. For example, a new preceptor who had not yet started the preceptor orientation training may not have an account in the learning management software or a recently moved preceptor may not yet have been removed from the learning management software. Within the clerkship administrative dataset, there were some cases with incomplete data on one or more variables. Given that most other variables were present, these cases were included despite certain missing variables. These variables were not imputed. The exclusion criteria were non-pharmacy preceptors, pharmacy resident preceptors, and international pharmacy preceptors. 

It was common for many preceptors to practice at more than one experiential site (e.g., both a general medicine and surgery unit at the same hospital). To avoid repetition, the dataset was further condensed to make each preceptor a separate case. To account for multiple sites that a preceptor could be associated with, a variable was added for the number of sites for each preceptor. The filtered and condensed dataset consisted of 1200 pharmacy preceptor cases. 

The following variables were included: gender, a graduate or professional degree in addition to a Bachelors or Doctorate of Pharmacy, completion of residency, practice setting, number of experiential sites, and time since pharmacist license date (0–5 years, 6–10 years, 11–20 years, and >20 years). The variables of graduate degree and residency completion were considered as separate variables and were not mutually exclusive. For example, a preceptor who completed a residency and a graduate degree were coded positive for both variables. Practice settings included ambulatory care, hospital pharmacy, outpatient pharmacy, administration, and other. Administration included sites related to management, managed care, drug information, and informatics. Examples of sites included in other included academia, compounding, corrections, hospice, industry, long-term care, and the state professional society. The numbers of advanced pharmacy practice experience (APPE) and introductory pharmacy practice experience (IPPE) students precepted in the 2019–2020 experiential year by the pharmacy preceptor were also determined. To build the final working dataset, the CPD completion database, indicating whether the pharmacy preceptors completed the CPD program, was combined with the preceptor database. 

### 2.2. Data Analysis

Descriptive statistics were performed for all variables by CPD program completion. Fischer’s exact test and Wilcoxon rank-sum test was performed for categorical variables and continuous or ordinal variables, respectively, by preceptors’ CPD program completion status. Logistic regression was performed to understand the likelihood of preceptors to complete the CPD program accounting for other variables. An alpha level of 0.05 was used for an indication of statistical significance. All data management and analysis were performed using Stata v 15.0 (StataCorp, Lakeway Drive College Station, Texas, TX, USA). 

### 2.3. Ethic Statement

This evaluation was certified as a quality improvement project by the UW-Madison Education and Social/Behavioral Sciences Institutional Review Board (certified 12 June 2019, Project Number 2019-0735).

## 3. Results

Of the 1200 preceptors included in the analysis, a majority (63%) were female (Table 1). All preceptors had a PharmD or a Bachelor of Pharmacy degree. Additionally, 45% of preceptors had completed a residency program but only 7% earned one or more graduate degree(s) in addition to their pharmacy licensure. Many of the preceptors (37%) had a practice experience of 11–20 years, whereas the smallest proportion of preceptors (12%) had less than five years of experience. 

Out of 1200 preceptors, 473 (39%) completed the CPD program (Table 2). Almost half of female preceptors (43%) completed the CPD program and were more likely to complete the CPD program than male preceptors (10% absolute increase, *p* < 0.001). Considering preceptors’ academic background, residency-trained preceptors were more likely to complete the CPD program when compared with preceptors without a residency (*p* = 0.01). However, preceptors with graduate degrees were less likely to complete the CPD training than those without an additional degree (26% and 40% respectively, *p* = 0.01). Time since initial licensure was not statistically significantly different (*p* = 0.12). Ambulatory care and hospital pharmacist preceptors were more likely to complete the CPD program than outpatient or administrative pharmacist preceptors. There was a higher median for number of APPE students precepted in the last year among preceptors who completed the CPD program (four students) compared with those who did not (two students; *p* < 0.001). Preceptors with more IPPE students were also more likely to complete the CPD program (*p* < 0.001). The results of the logistic regression were consistent with the primary analysis (Table 3).

## 4. Discussion

The results suggest that the tendency for preceptors to complete CPD programs varies, and may depend on several preceptor characteristics. Female preceptors, residency-trained preceptors without a graduate degree, ambulatory care and hospital pharmacist preceptors, and preceptors with a higher number of students were variables which indicated a higher likelihood to complete the CPD program. The results of this evaluation add to the existing literature on the barriers and facilitators to CPD uptake by preceptors and pharmacists. For instance, it was predictable that residency-trained preceptors were more likely to complete the CPD training given that the American Society of Health-System Pharmacists residency standards require residents to create professional development plans and have them reviewed quarterly [9]. The residency-trained preceptors may have had pre-existing buy-in and familiarity with the CPD process. This was also consistent with practice setting as pharmacists in ambulatory care and hospital pharmacy settings are more likely to have completed a residency [10]. In order to encourage further engagement of non-residency-trained preceptors, it may be helpful to tailor the advertisements of the training. This could include a brief overview of the stages of CPD as well as a statement of some of the benefits, including the ability to develop a personalized plan. These advertisement strategies would allow preceptors to see what the activity will entail and how it could benefit them personally. 

Preceptors who had more APPE and IPPE students were also more likely to complete the CPD program. These preceptors may have felt the CPD process with a focus on improvements in clinical teaching was more relevant and worth their time, given they are using these skills more often. However, the students who sign up for sites that only accept 1–2 students per year are still expected to receive a high-quality experience with an informed preceptor. The CPD program in the original evaluation was advertised during the start of the experiential education year; however, some preceptors may not have students until the last block. In order to encourage preceptors who have limited students to partake in the CPD process, the CPD program should be advertised year-round to allow preceptors to complete the training when it is most relevant to their precepting schedule, perhaps including an email reminder at the start of each experiential block. 

There was a non-statistically significant trend in decreased participation in the CPD process as time since licensure increased. While years practicing pharmacy does not necessarily correlate to number of years as a preceptor, it raises the question of whether more experienced pharmacists believe they are already capable preceptors and do not feel compelled to complete precepting-focused CPD. This suggestion is supported by weak evidence that older practitioners may be less likely to participate in CPD activities [11]. To the authors’ knowledge, no previous studies have attempted to correlate years in practice with the age or quality of a preceptor. However, studies have shown that students perceive the intangible skills in teaching and mentoring as characteristics of quality preceptors [12,13,14,15]. These skills are not necessarily learned simply through more time being a pharmacist or even passively with more time as a preceptor. Marketing preceptor-focused CPD activities towards experienced pharmacists may require an emphasis on the skills that are the focus of preceptor development along with encouragement to self-reflect on their current skillsets in precepting. 

There are several strategies that could be employed to increase overall preceptor uptake of a CPD program, regardless of preceptor characteristics. Offering CE credits for CPD activities seemed to be effective. Ninety percent of the preceptors who completed the original CPD program claimed CE credit [7]. The option to claim these general CE credits may have motivated preceptors to complete the training, leading to a higher response rate. Future CPD programs should continue to offer CE credit as past studies have shown pharmacists are motivated by needing to complete CE for licensure requirements [16,17]. At least one known health system made the original CPD program a requirement for its pharmacist preceptors [7]. Of the 175 preceptors from this health system, 68% completed the CPD program compared with 35% of preceptors with other employers (*p* < 0.001). It should be known that the distribution of preceptors from this health system may have influenced the results of the completion of the CPD program by practice setting. Schools of pharmacy could communicate with the health systems that provide the largest number of preceptors and discuss requiring its preceptors to complete the training and utilize the CPD process. This can help increase training of preceptors and potentially eliminate or decrease the time barrier, which would be valuable as pharmacists have consistently identified time as a barrier to completing CE and CPD activities [16,17,18]. Health systems who require CPD training (with associated CE) could also provide time for preceptors to complete the training while on duty or provide incentive for time spent outside of scheduled hours. Health systems stand to glean several benefits from having pharmacy preceptors complete the CPD process. By improving the quality of pharmacy education provided by its preceptors, a health system is investing in future pharmacists, who may end up being potential job candidates at that institution. As suggested by Tofade et al., preceptor CPD training could also be incorporated into already existing infrastructure of in-person annual preceptor training or workshops [6]. However, the preceptors who were most likely to complete the online CPD training may also be more likely to attend other in-person training offerings, perhaps indicative of their internal motivations to improve their precepting. 

This evaluation is not without limitations. When considering completion of a residency or graduate degree, if a preceptor did not have that information completed, it was considered missing data in this analysis. There are also factors related to motivation to improve as a preceptor that were not measured. For example, some preceptors are motivated and choose to be a preceptor while some may be a preceptor given the culture of teaching at their place of employment. While we would expect those with internal motivation and interest in precepting to be more likely to complete a CPD program, this was not quantified in this analysis. Motivation, however, can be quantified by utilizing the self-determination theory, as several studies in the Netherlands have done to assess the relationship between motivation and CE/CPD completion among Dutch pharmacists [19,20,21]. Tjin A Tsoi and colleagues showed that relative autonomous motivation, which is a measure of a person’s internal motivation corrected for external motivators, had a positive correlation with completion of CE hours [19]. Pharmacists may also have different motivational profiles that affect the quantity and quality of the CE/CPD they complete and these profiles appear to be associated with factors such as gender, practice setting, and type of or place in pharmacist training [21]. Collectively, these findings may help explain the variations in the uptake of CPD reported in this study and help support the notion that targeting CPD towards participant characteristics may improve participation. However, further studies applying the self-determination theory to this subset of pharmacists are necessary to draw more definitive conclusions.

Monitoring changes in participation rates after adopting the suggested advertising techniques is warranted. Other future directions include follow-up from the initial CPD program to determine the frequency of preceptor completion of learning plans as well as incorporation of behavior changes into their clinical teaching. While past literature suggests there can be an initial improvement in behavior changes [3], longer-term studies have suggested that the efficacy of CPD programs can wane over several years [22]. This suggests the need for follow-up and refresher CPD programs, which the UW-Madison SOP has incorporated into the preceptor development materials and will continue to determine the impact of in the future.

## 5. Conclusions

This evaluation found that female preceptors, residency-trained preceptors, ambulatory care and hospital pharmacist preceptors, and preceptors with a higher number of students were more likely to complete a preceptor CPD program. In addition to identifying characteristics of the preceptors who completed the CPD program, a number of potential suggestions to improve the rate of preceptor completion of the CPD program were described. Future directions include monitoring changes in the CPD program participation rates following changes in advertisements, evaluating preceptors’ motivation towards completion of the CPD program, and exploring collaborations with employers.

## Figures and Tables

**Table 1 pharmacy-08-00121-t001:** Characteristics of pharmacy preceptors.

Variable	Frequency N(%)	Mean (±sd)	Range
Gender			
Male	445 (37)		
Female	755 (63)		
Licensed since (years)			
0–5	149 (12)		
6–10	309 (26)		
11–20	439 (37)		
>20	303 (25)		
Graduate degree (n = 1198)			
Yes	427 (45)		
No	517 (55)		
Residency (n = 944)			
Yes	81 (7)		
No	1117 (93)		
CPD Completion			
Yes	473 (39)		
No	727 (61)		
Practice Setting			
Ambulatory care	92 (8)		
Hospital pharmacy	490 (41)		
Outpatient pharmacy	380 (32)		
Administration	107 (9)		
Other	131 (11)		
Number of sites^1^		1.43 (1.5)	1–42
Number of APPE students^1^		4.25 (5.3)	0–72
Number of IPPE students^1^		3.1 (5.4)	0–32

^1^ Several preceptors were linked to all sites at a health center for administrative purposes. In those cases, they were unlikely to directly interact with all students who were at those experiential sites. APPE = advanced pharmacy practice experience; CPD = continuing professional development; IPPE = introductory pharmacy practice experience.

**Table 2 pharmacy-08-00121-t002:** Frequency of pharmacy preceptors who completed continuing professional development (CPD) training.

Variable	Frequency	CPD Completion N(%)	*p*-Value
Gender			<0.001
Male	445	145 (33)	
Female	755	328 (43)	
Licensed since (years)			0.12
0–5	149	66 (44)	
6–10	309	132 (43)	
11–20	439	170 (39)	
>20	303	105 (35)	
Graduate degree (n = 1198)			0.01
Yes	81	21 (26)	
No	1117	452 (40)	
Residency (n = 944)			0.01
Yes	427	192 (45)	
No	517	190 (37)	
Practice setting			0.01
Ambulatory care	92	46 (50)	
Hospital pharmacy	490	208 (42)	
Outpatient pharmacy	380	125 (33)	
Administration	107	34 (32)	
Other	131	60 (46)	

CPD = continuing professional development.

**Table 3 pharmacy-08-00121-t003:** Logistic regression for preceptor CPD completion.

Variable	Odds Ratio	*p*-Value
Female gender	1.38	0.023
Practice experience	0.94	0.39
Graduate degree	0.45	0.008
Residency	1.37	0.24

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
