# Peer review of "Characteristics of Individuals Who Chose to Participate in a Preceptor Continuing Professional Development Program"

_pharmacy, 2020, doi:10.3390/pharmacy8030121_

Round 1

Reviewer 1 Report

  • Please change the title to "Continuing" (not "Continuous") as this is the term you have used throughout the article, and it the more commonly adopted term for CPD.
  • Lines 29-30: The four steps of CPD that you have described in the opening sentence of the Introduction have been replaced by ACPE with five stages in an "infinity" cycle (now including "Apply"), and with "Record and Review" as a central element depicted within the right-hand learning cycle. Unfortunately, the CPD Overview that you have chosen to cite is very much in need of updating. There are better resources on the ACPE website for you to cite and use for your definition and description of the CPD model. Please change this important first sentence in the manuscript.
  • ACPE in References 1 and 5 is "Accreditation" not "American."
  • In Discussion, you refer several times to motivation, including "internal motivations." I agree with you that this is likely a key issue, if not the key issue when it comes to uptake of the CPD model. I am sure that internal motivation has a strong correlation with what you found to significantly impact participation the CPD activity (e.g., residency training, number of students precepted). You correctly point out that motivation is difficult to measure, but there are now some published studies looking at intrinsic and extrinsic motivators, and how they impact adoption of the CPD approach. I would suggest that you have a quick look at the literature and include some of the findings and conclusions (and references) in your Discussion. (Examples: (1) Sharon L N M Tjin A Tsoi 1, Anthonius de Boer, Gerda Croiset, Andries S Koster, Rashmi A Kusurkar. Factors Influencing Participation in Continuing Professional Development: A Focus on Motivation Among Pharmacists. (2) Sharon L. Tjin A Tsoi, Anthonius de Boer, Gerda Croiset, Rashmi A. Kusurkar, and Andries S. Koster.A Longitudinal Approach to Changes in the Motivation of Dutch Pharmacists in the Current Continuing Education System.)
  • With regard to motivation, this is something that I would encourage the authors to explore in future studies. I am not convinced that just changing advertising strategies will have the desired result (although it can certainly be studied as well).
  • Lines 43 & 48: I am not sure that a 35% completion rate is a "positive" response rate, but that's an opinion. I do agree, however, that the percentage who expressed that the program had perceived value was a "positive" result.
  • Line 110: I suggest dropping the word "equivalent."
  • Line 112: 37% is not a "majority;" please re-state.
  • Line 149: Please fix the wording.
  • Line 163: Number of years in practice - as this generally correlates with age, you might want to cite some other studies that have explored this issue. I believe there is evidence that older pharmacists are less inclined to participate in CPD activities.
  • Line 208: Application of learning is arguably one of the most important elements of the CPD model, so incorporation of behavior changes is definitely something that should be documented and evaluated.
  • Line 211: There is no doubt that adopting a CPD approach requires more effort and commitment by the learner; maintaining motivation is therefore critically important, so I agree that follow-up is needed; however, seeing the impact of one's learning can also be a strong motivator (See Rouse, Trewet, Janke. Advancing learning to advance pharmacy practice. JAPhA.)
  • I would suggest adding a Conclusion.

Author Response

  • Please change the title to "Continuing" (not "Continuous") as this is the term you have used throughout the article, and it the more commonly adopted term for CPD.

Thank you for pointing this out. We made this change.

  • Lines 29-30: The four steps of CPD that you have described in the opening sentence of the Introduction have been replaced by ACPE with five stages in an "infinity" cycle (now including "Apply"), and with "Record and Review" as a central element depicted within the right-hand learning cycle. Unfortunately, the CPD Overview that you have chosen to cite is very much in need of updating. There are better resources on the ACPE website for you to cite and use for your definition and description of the CPD model. Please change this important first sentence in the manuscript.

The CPD cycle we are familiar with is the infinity cycle described in the peer review feedback. The feedback made it clear we had not described it well. We adjusted the description to better describe ACPE’s CPD model. We believe that the citation is fine as it is the webpage to all of ACPE’s CPD resources.

  • ACPE in References 1 and 5 is "Accreditation" not "American."

Thank you for pointing this out. We made this change.

  • In Discussion, you refer several times to motivation, including "internal motivations." I agree with you that this is likely a key issue, if not the key issue when it comes to uptake of the CPD model. I am sure that internal motivation has a strong correlation with what you found to significantly impact participation the CPD activity (e.g., residency training, number of students precepted). You correctly point out that motivation is difficult to measure, but there are now some published studies looking at intrinsic and extrinsic motivators, and how they impact adoption of the CPD approach. I would suggest that you have a quick look at the literature and include some of the findings and conclusions (and references) in your Discussion. (Examples: (1) Sharon L N M Tjin A Tsoi 1, Anthonius de Boer, Gerda Croiset, Andries S Koster, Rashmi A Kusurkar. Factors Influencing Participation in Continuing Professional Development: A Focus on Motivation Among Pharmacists. (2) Sharon L. Tjin A Tsoi, Anthonius de Boer, Gerda Croiset, Rashmi A. Kusurkar, and Andries S. Koster.A Longitudinal Approach to Changes in the Motivation of Dutch Pharmacists in the Current Continuing Education System.)

Thank you for the suggested resources. We have reviewed the referenced and revised that section of the discussion.

  • With regard to motivation, this is something that I would encourage the authors to explore in future studies. I am not convinced that just changing advertising strategies will have the desired result (although it can certainly be studied as well).

Thank you for the suggestion. The current work on this project has incorporated the theory of planned behavior which I think will better capture this element.

  • Lines 43 & 48: I am not sure that a 35% completion rate is a "positive" response rate, but that's an opinion. I do agree, however, that the percentage who expressed that the program had perceived value was a "positive" result.

We agree a 35% response rate is not ideal – we were commenting more on the high number of 491 preceptors who completed the program. This is much higher than similar other preceptor development programs we have reviewed in the literature. We did remove “rate” from the sentence in light of a 35% response being low.

  • Line 110: I suggest dropping the word "equivalent."
    We made this change

  • Line 112: 37% is not a "majority;" please re-state.
    We made this change

  • Line 149: Please fix the wording.

We made this change

  • Line 163: Number of years in practice - as this generally correlates with age, you might want to cite some other studies that have explored this issue. I believe there is evidence that older pharmacists are less inclined to participate in CPD activities.

We found some weak evidence that suggests there may be a link between age and participation in CPD. We added a comment related to this potential trend. As we only found weak evidence and given our limitation in age and the data we have for years in practice, we did not wish to further discuss this trend.

  • Line 208: Application of learning is arguably one of the most important elements of the CPD model, so incorporation of behavior changes is definitely something that should be documented and evaluated.
    This is a future direction that we are actively working on – as stated in the future directions section of the manuscript. We expect to have these results later this year!

  • Line 211: There is no doubt that adopting a CPD approach requires more effort and commitment by the learner; maintaining motivation is therefore critically important, so I agree that follow-up is needed; however, seeing the impact of one's learning can also be a strong motivator (See Rouse, Trewet, Janke. Advancing learning to advance pharmacy practice. JAPhA.)

We will look into this in our future work. Thank you for the suggested reference.

  • I would suggest adding a Conclusion.
    We have added this.

Reviewer 2 Report

Thank you for submitting this paper.  I think the topic is very interesting.  I do have a couple of suggestions.

  1. The introduction needs to include more background information about the CPD program that was offered.  I see the information in the methods but I think the intro needs to include a general background description of the program offered before going into numbers of preceptors that completed the program.
  2. Need to include other evidence to support the lack of preceptor engagement in CPD other than the previous study conducted by this group.
  3. Might be good to include some rationale behind how you selected the variables you included in your analysis.  I am curious why practice setting wasn't a variable that was included.
  4. It is not clear if you are including residency training in "graduate degree" or if this is separate.  This should be clarified.
  5. Table 2 does not include IPPE and APPE student numbers but this was presented in the text as significant.  I would include in table.
  6. As you mentioned in limitations that one site did require the training, is it possible to analyze the data without those individuals to see if there was an impact?  Or at least disclose the number of individuals that were impacted to see % of total sample size.

Author Response

  1. The introduction needs to include more background information about the CPD program that was offered.  I see the information in the methods but I think the intro needs to include a general background description of the program offered before going into numbers of preceptors that completed the program.

We made some revisions to better introduce the program before the initial results from the CPD program.

  1. Need to include other evidence to support the lack of preceptor engagement in CPD other than the previous study conducted by this group.

We added some other examples

  1. Might be good to include some rationale behind how you selected the variables you included in your analysis.  I am curious why practice setting wasn't a variable that was included.

Thank you for the suggestion. We were able to go back and include practice setting in the analysis.

  1. It is not clear if you are including residency training in "graduate degree" or if this is separate.  This should be clarified.
    They are separate variables. We have further clarified the language in the methods section.

  2. Table 2 does not include IPPE and APPE student numbers but this was presented in the text as significant.  I would include in table.

Table 2 displays results from the Fischer’s exact test for categorical variables only. APPE and IPPE scores are continuous variables and the Wilcoxon Rank-sum test generates only medians and p values. These results are thus only mentioned in the narrative section and not in Table 2.

  1. As you mentioned in limitations that one site did require the training, is it possible to analyze the data without those individuals to see if there was an impact?  Or at least disclose the number of individuals that were impacted to see % of total sample size.

Yes, we were able to go back and determine this. We have added a comment on this in the discussion section.

Reviewer 3 Report

Very interesting research! 

However, I am slightly confused in terms of your objective. Your title states that you are looking to find characteristics of those who did the CPD program, however the objective states that you want to identify characteristics of those who did not participate and then in the data collection and management section you state "for the present analysis to determine characteristics of preceptors who completed the CPD program."

This is slightly confusing so I was wondering if you would be able to provide more clarification in this area. Are you trying to determine the characteristics of those who did not complete the survey based on the direct inverse of those who did? 

I did like the strategies that you included in your discussion regarding the potential ways to get different preceptor engaged, they seem very tailored to the groups you identified. Your future plans are also very sound and appropriate. 

Author Response

Your feedback was a correct interpretation of our work. We did focus on the characteristics of who did complete the training to consider how to better target those who did not complete the training. We have made some adjustments to the manuscript to better describe this. Thank you for the suggestion.